# TAIL: Task-specific Adapters for Imitation Learning with Large Pretrained Models

**Abstract:** The full potential of large pretrained models remains largely untapped in control domains like robotics. This is mainly because of the scarcity of data and the computational challenges associated with training or fine-tuning these large models for such applications. Prior work mainly emphasizes effective *pretraining* of large models for decision-making, with little exploration into how to perform data-efficient continual *adaptation* of these models for new tasks. Recognizing these constraints, we introduce TAIL (**T**ask-specific **A**dapters for **I**mitation **L**earning), a framework for efficient adaptation to new control tasks. Inspired by recent advancements in parameter-efficient fine-tuning in language domains, we explore efficient fine-tuning techniques—e.g., Bottleneck Adapters, P-Tuning, and Low-Rank Adaptation (LoRA)—in TAIL to adapt large pretrained models for new tasks with limited demonstration data. Our extensive experiments comparing prevalent parameter-efficient fine-tuning techniques and adaptation baselines suggest that TAIL with LoRA can achieve the best post-adaptation performance with only 1% of the trainable parameters of full fine-tuning, while avoiding catastrophic forgetting and preserving adaptation plasticity in continual learning settings.

**Keywords:** Adaptation of Pretrained Model, Continual Imitation Learning

## 1 Introduction

A desired property of an autonomous agent is the ability to adapt efficiently to novel tasks. In vision and language domains, large pretrained models have demonstrated adaptation to new tasks with just a few examples through prior knowledge obtained from internet-scale datasets [1, 2, 3]. Similar methods have also been applied in decision-making and control applications [4, 5, 6]. However, new control tasks are more difficult to adapt to than the aforementioned vision and language domains due to (1) the lack of internet-scale control data and (2) how optimal actions can vary significantly from task-to-task, even under shared observation spaces. As such, these large-scale decision-making models still rely on a close alignment between training and testing tasks.

In contrast, agents deployed in challenging environments need to adapt to major task variations—take, for example, a general household robot. Equipped with a factory-pretrained policy, the robot will be employed in unique ways by every household. Thus, the robot will need to *continually adapt* in order to best serve each one, e.g., by fine-tuning its capabilities on a few demonstrations. Because most prior decision-making papers adapt to new tasks by fine-tuning the entire model [7, 8, 9, 10, 11, 12, 6, 13], mastering each new skill requires great computational cost and often leads to catastrophic forgetting of old ones. An alternative approach would be to store a separate policy per new task, which leads to unreasonable storage spaces. What would be the best way for agents to *efficiently adapt* to a stream of novel tasks without having to trade off computation, storage, and performance on older tasks?

To answer this question, we propose **T**ask-specific **A**dapters for **I**mitation **L**earning, shown in Fig. 1, a framework for efficient adaptation to new control tasks. Through TAIL we (1) effectively incorpo-

Submitted to the 7th Conference on Robot Learning (CoRL 2023). Do not distribute.

rate lightweight adapter modules into pretrained decision-making models and (2) comprehensively compare efficient adaptation techniques implemented in TAIL in a continual imitation learning setting. Notably, we examine parameter-efficient adaptation techniques (PEFT) used for large language models; we explore the potential of adapters [14], prefix tuning [15], and low-rank adaptation (LoRA) [16] in fostering efficient and continual adaptation in large pretrained decision-making models. These works stand out as they introduce a small number of *new* parameters which help: avoid catastrophic forgetting, maintain training plasticity for continual learning, avoid overfitting with limited adaptation data, and reduce computational and memory burden. Investigating these works in control and continual learning setup is important because, unlike in language domains, test losses are often not proportional to the task performance [17, 18]—efficient adaptation insights from language models may not transfer to decision-making ones. Thus, investigation of these adaptation techniques for decision-making is crucial for deploying continually adapting agents in the real world.

We compare PEFT techniques implemented in TAIL against commonly used adaptation methods in the imitation learning literature. In our experiments, we discover that TAIL with LoRA leads to the best post-adaptation performance as it learns additional low-rank weight matrices for a specific task, allowing it to preserve the original pretrained representations while being resilient against overfitting in the limited-data regime. These capabilities are especially important for agents operating in new, challenging environments, such as the aforementioned general household robots. Our analysis also reveals important insights into the strengths and limitations of each adaptation strategy. Instead of performing full fine-tuning of the entire model, TAIL only introduces a small number of additional parameters and exclusively updates these new parameters without making changes to the original model. These additional parameters make up a mere $1.17\%$ of the size of the original model. Importantly, this results in approximately $23\%$ less GPU memory consumption, to achieve $22\%$ higher forward adaptation success rate than full fine-tuning, while avoiding catastrophic forgetting. Notably, these results are contrary to many results from the vision and language model literature which show that full fine-tuning works better [19, 20, 21].

In summary, this work bridges a crucial gap in research into efficient and continual adaptation for pretrained decision models by introducing a framework for continual imitation learning, TAIL, and thoroughly analyzing the effects of different efficient adaptation methods to inform future research. Our comprehensive results clearly show the effectiveness and practicality of our proposed approach.

## 2 Related Work

**Pretrained Models for Control.** Researchers have long studied the use of pretrained models for better downstream transfer to related tasks [22, 23, 24]. Recent works have examined using the representations learned by pretrained visual models for control [25, 26, 27, 28, 29]. These methods either do not attempt adaptation to new tasks, or perform expensive full-fine-tuning for adaptation. In contrast, our method, TAIL, is a framework for efficient adaptation of decision-making models.

**Parameter-Efficient Fine-Tuning (PEFT) Techniques.** PEFT has gained traction as a way to adapt large pretrained models without significantly increasing parameters. Techniques such as adapters [14], LoRA [16], and prompt tuning [15] incorporate lightweight modules or continuous prompts optimized for downstream tasks, all while preserving the original model weights. Liang et al. [30], Sharma et al. [31] propose the use of adapters in robotics settings, but they do not examine other PEFT techniques and focus on adaptation to a single task suite. We instead examine the performance of various PEFT techniques implemented with TAIL in a realistic *continual learning* scenario and demonstrate in Sec. 5 that TAIL works better than RoboAdapter [31] in this setting.

**Continual Learning.** Continual learning agents should be able to transfer knowledge (e.g., by continually fine-tuning) or experience (e.g., training data) from previously learned tasks to new tasks [32, 33, 34, 35]. However, with large pretrained models trained on large datasets, fine-tuning the entire model is computationally costly yet risks catastrophic forgetting, and transferring training data from other tasks is too memory inefficient in the face of a large stream of new tasks. Therefore, we present a study into efficient fine-tuning techniques which, when integrated with TAIL, can help inform future research of continual learning.

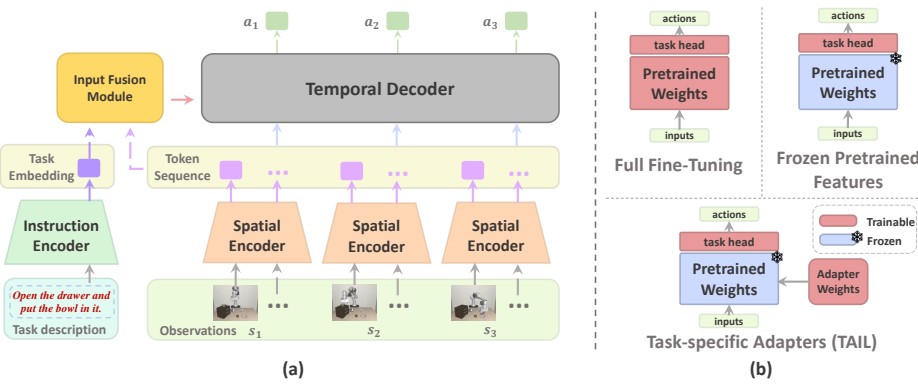

Figure 1: **(a)**: The multi-modal, transformer policy architecture we utilize for pretraining. We encode language task descriptions with a pretrained CLIP instruction encoder and image observations with a pretrained CLIP spatial encoder. We additionally encode state observations (not pictured) which, along with the observation embeddings, are embedded into a sequence of tokens used by the temporal decoder transformer to predict single-step action distributions. We include an input fusion module to explicitly combine the task embedding with the observation token sequence for better instruction-following ability. **(b)**: The three types of fine-tuning paradigms we test, with TAIL at the bottom right. For further architecture details, see Appendix Sec. A.

## 3 Preliminaries

### 3.1 Continual Imitation Learning

The agent encounters a sequence of $K$ tasks, denoted as $\{\mathcal{T}_1, \ldots, \mathcal{T}_K\}$. Each task $\mathcal{T}_k = (\mu_k^0, g_k)$ is characterized by an initial state distribution $\mu_k^0$ and a goal predicate $g_k$. Goals for tasks can be specified using language instructions, providing clear context [36, 12]. For every task $\mathcal{T}_k$, the agent receives $N$ demonstration trajectories $\mathcal{D}_k = \{\tau_k^1, \ldots, \tau_k^N\}$. In this paper, we use the standard behavioral cloning loss to optimize the agent's policy $\pi$ over these demonstrations, however we note that TAIL can be used with other training objectives as well:

$$\hat{\boldsymbol{\theta}} = \min_{\boldsymbol{\theta}} \sum_{k=1}^{K} \mathop{\mathbb{E}}_{s_t, a_t \sim \mathcal{D}_k} \left[ \sum_{t=0}^{l_k} \mathcal{L}\left(\pi(a|s_{\leq t}, \mathcal{T}_k; \boldsymbol{\theta}), a_k^t\right) \right]. \tag{1}$$

Here, $\mathcal{L}$ is a supervised action prediction (e.g., mean squared error or negative log likelihood) loss, $l_k$ is the length of demonstrations for task $\mathcal{T}_k$, and $\boldsymbol{\theta}$ refers to the *learnable parameters* of the network. Notably, after learning task $\mathcal{T}_k$, the agent cannot access *additional* data from preceding tasks. This presents a continual learning challenge, emphasizing the importance of transferring knowledge across tasks without the risk of catastrophic forgetting [37].

### 3.2 Pretrained Decision-Making Models

Here, we briefly describe common features of large pretrained decision-making model architectures used for embodied agents. We incorporate key components shared amongst these models into the architecture of the model that we pretrain to evaluate efficient adaptation, pictured in Fig. 1(a).

**Transformer Backbone.** Most recent work training large-scale decision-making models [4, 38, 6] utilize a transformer backbone [39] that attends to tokenized observations from prior timesteps. We adopt a standard GPT-2 [40] transformer decoder (Fig. 1(a), temporal decoder) with separate encoders for each input modality and continuous action distribution outputs.

**Pretrained Input Encoders.** Encoders pretrained on large, diverse datasets can produce rich, well-structured embeddings which make it easier to learn the downstream tasks [36, 4]. Therefore, we utilize pretrained CLIP image and textual encoders [2].

**Input Modality Fusion.** The idea of explicitly "fusing" different input modalities has seen great success not only in domains like vision and language [41], but also in agent learning [36, 4]. Similarly, we utilize FiLM layers [41] (Fig. 1(a), input fusion module) to fuse language task specifications with observations.

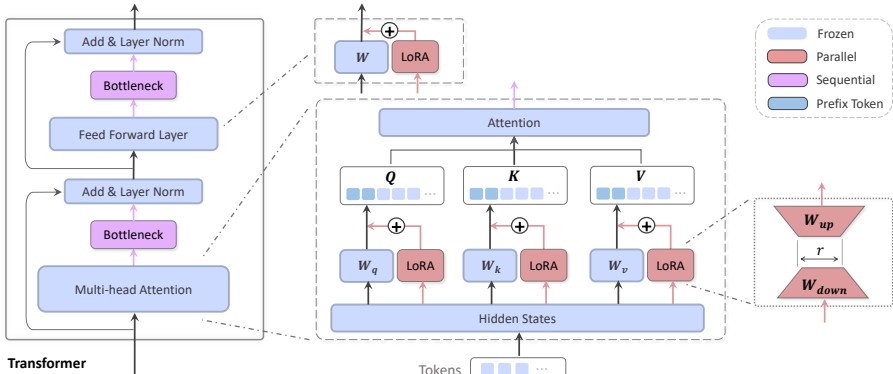

Figure 2: Demonstration of three weight integration styles of TAIL for a Transformer block: sequential (bottleneck adapter), parallel (LoRA), and prefix token (prefix/prompt-tuning).

## 3.3 Adapting pretrained models for new tasks

One standard adaptation method in prior research is full fine-tuning (FFT) of all model parameters (Fig 1(b), top left). Though straightforward, it is resource-intensive and prone to overfitting with limited data [11]. There is also a risk of distorting pretrained features, resulting in the loss of prior tasks—a phenomenon known as **catastrophic forgetting** [37]. Evidence also suggests that extensive fine-tuning might undermine a model's rapid adaptability to new tasks, an effect referred to as the loss of **model plasticity and capacity** [42, 43, 44]. Such issues become more prominent in continual learning contexts [32]. Another standard adaptation method is the use of frozen pretrained features (FPF, Fig 1(b) top right). FPF ensures the retention of knowledge acquired from previous tasks by tuning a task-specific head. However, as noted in Sharma et al. [31], it is not expressive enough for out-of-distribution or especially complex tasks. Given these challenges, there's a clear need for a more advanced fine-tuning paradigm that addresses catastrophic forgetting while maintaining model plasticity for adapting to new tasks, all in a data and computationally resource-efficient manner.

## 4 Task-specific adapters for imitation learning

In this section, we outline how we perform efficient adaptation on pretrained models through our **T**ask-specific **A**dapters for **I**mitation **L**earning framework, depicted in Fig 1(b). Different from the FPF approach which simply substitutes the policy head for every new task, TAIL introduces a small set of new weights, serving as a lightweight plugin to address specific tasks.

### 4.1 Adapter Weights Integration

The concept of an adapter can be best conceptualized as a modular plugin to the base model, customized for specific downstream tasks, that does not affect the model's pretrained representations. We mainly explore three prevalent styles of integration for TAIL: **Parallel** [16], **Sequential** [14, 31], and **Prefix Token** [15, 45, 46], all of which are showcased with a Transformer block in Fig. 2. Parallel and sequential integration techniques are generally applicable to any model with feedforward layers, while the prefix token style method is especially tailored for Transformers.

Given a pretrained model, let's consider *one* layer weight matrix in it, denoted as $W \in \mathbb{R}^{d \times k}$. Its input and output hidden states are $h_{in} \in \mathbb{R}^d$ and $h_{out} \in \mathbb{R}^k$, respectively. We have $h_{out} = W^\top h_{in}$. Next, we detail how to apply parallel and sequential insertions to the pretrained weight matrix.

**Parallel Integration (LoRA).** This integration method, often associated with Low-Rank Adaptation (LoRA) [16], introduces trainable low-rank matrices $W_{down} \in \mathbb{R}^{d \times r}$ and $W_{up} \in \mathbb{R}^{r \times k}$. Here, $r \ll \min(d, k)$ represents the rank and is usually much smaller than the dimensions of the original matrix. These matrices are typically integrated in parallel with the original weight matrix $W$ through addition, as shown as LoRA in Fig. 2:

$$h_{out} = W^\top h_{in} + \alpha W_{up}^\top W_{down}^\top h_{in}, \qquad (2)$$

with $\alpha$ being a hyperparameter to modulate task-specific adjustments. The above equation can also be formulated as: $h_{out} = (\boldsymbol{W} + \alpha \boldsymbol{W}_{down}\boldsymbol{W}_{up})^\top h_{in} = (\boldsymbol{W} + \alpha \Delta \boldsymbol{W})^\top h_{in}$, where $\Delta \boldsymbol{W}$ denotes the weight modifications for new tasks, and thus the columns of $\boldsymbol{W}_{down}$ and $\boldsymbol{W}_{up}$ can be interpreted as a new basis that contains task-specific knowledge. As observed by Aghajanyan et al. [47], despite projecting to a condensed subspace with small "intrinsic dimensions," pretrained models can still learn effectively. By introducing the two low-rank matrices, the original weight matrices $\boldsymbol{W}$ can be adeptly tailored with a minimal increase in parameters. Though LoRA was originally crafted for large language models—specifically for the query and value projections matrices $W_Q$ and $W_V$ in multi-head attention [16]—it is easily applied to other linear layers as well, such as the Transformer's feedforward layers [21].

**Sequential Integration (Bottleneck Adapter).** Renowned in the language model domain, the Bottleneck Adapter introduces bottleneck layers within the model [14, 31] by appending a trainable bottleneck layer after the feedforward network in each Transformer layer. Similar to LoRA, this bottleneck consists of down and up projections, $\boldsymbol{W}_{down}$ and $\boldsymbol{W}_{up}$, which first shrink then restore the dimensions of token hidden states. Formally, for the feedforward network's input $h_{in}$ and a bottleneck size $r$, the output $h_{out}$ is:

$$h_{out} = \boldsymbol{W}_{up}^\top \phi \left( \boldsymbol{W}_{down}^\top (\boldsymbol{W}^\top h_{in}) \right), \tag{3}$$

where $\phi$ denotes a nonlinear activation function. The Bottleneck Adapter (Fig. 2) acts as a filter, isolating relevant information for specific tasks. Yet, filtering often requires a larger bottleneck size compared to that of LoRA, leading to more parameters. Additionally, the sequential insertion can increase latency compared to the parallel nature of LoRA [16].

**Prefix Token Integration (Prefix & Prompt-Tuning).** In this style, a set of learnable prefix tokens are appended or prepended to the input sequence [15, 45, 46]. Let's consider an input sequence $\mathbf{s} \in \mathbb{R}^{n \times d}$, where $n$ is the sequence length and $d$ is the embedding dimension. The prefix tokens can be represented as $\mathbf{p} \in \mathbb{R}^{m \times d}$, where $m$ denotes the number of prefix tokens. These vectors act like virtual tokens which the original tokens can attend to. They are initialized and learned during the task-specific adaptation phase. The modified input sequence, after appending the prefix tokens, can be expressed as $\mathbf{S} = [\mathbf{p}; \mathbf{s}] \in \mathbb{R}^{(m+n) \times d}$. The model then processes this extended sequence. These prefix tokens can be viewed as task descriptors that are designed to guide the model towards the desired task-specific behavior (see ■ in Fig. 2).

### 4.2 TAIL for continual imitation learning

We consider the continual imitation learning problem as a typical application of the proposed TAIL adaptation paradigm. The goal of continual imitation learning is to ensure that the model performs effectively on the current task and without significant degradation of performance in past tasks. Given pretrained model weights, denoted as $\boldsymbol{\theta}$, and a new task $\mathcal{T}_k$ with demonstrations $\mathcal{D}_k = \{\tau_k^1, \ldots, \tau_k^N\}$, we initialize the task-specific adapter weight $\boldsymbol{\omega}_k$ with far less parameters than the base model: $|\boldsymbol{\omega}_k| \lll |\boldsymbol{\theta}|$. The adapter weights are inserted into the model through the integration methods introduced in Sec. 4.1. By optimizing the behavior cloning loss in Eq. 1 w.r.t $\boldsymbol{\omega}_k$ while keeping the pretrained weights frozen, the policy adapts to $\mathcal{T}_k$ without interfering with previous tasks.

To execute a task, the corresponding lightweight adapters are loaded as a plugin of the pretrained network weights. For example, when revisiting a prior task $T_j$, where $j \leq k$, the model is configured to solely activate the $j$-th adapter $\boldsymbol{\omega}_j$. This entire procedure can be streamlined as follows:

1. For an incoming task $\mathcal{T}_k$, acquire the training set $\mathcal{D}_k$, initialize a task-specific adapter $\boldsymbol{\omega}_k$.

2. Combine adapter $\boldsymbol{\omega}_k$ with the base model $\boldsymbol{\theta}$ using either parallel, sequential, or prefix token.

3. Train the adapter on $\mathcal{D}_k$ to optimize Eq. 1 for $\boldsymbol{\omega}_k$, keeping pretrained parameters $\boldsymbol{\theta}$ frozen.

In essence, TAIL ensures task-specific knowledge is contained within the adapters, thereby enabling efficient adaptation without catastrophic forgetting. It's also worth noting that the TAIL framework is flexible. The choice of integration method or the specific architecture of the adapter can be tailored based on the complexity of the task or the available computational resources.

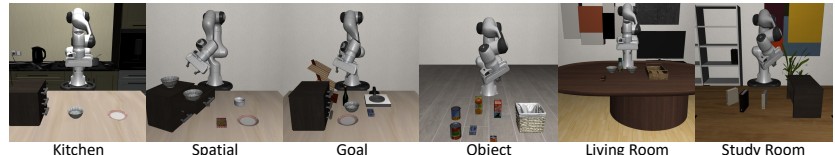

| Kitchen | Spatial | Goal | Object | Living Room | Study Room |

Figure 3: Our task suites for continual imitation learning (excluding LIBERO-10). The robot, placed in a tabletop environment, is equipped with a 6-DOF arm and a parallel gripper. It receives RGB images from two views, joint states, and language instructions, and is tasked with producing continuous actions to control its arm.

## 5  Experiments

### 5.1  Datasets and Benchmark Suites

We utilize the LIBERO robotic manipulation continual learning benchmark [48], which features a diverse range of tasks that mirror human daily activities. Each task is specified via natural language instructions. We craft a *pretraining* task suite, named **Kitchen**, involving 40 diverse tasks sourced from the LIBERO-90 dataset's kitchen scenes. We then evaluate *adaptation* to 5 separate task suites. The **Spatial** task contains the same objects in each scene but with different spatial layouts; each task in the **Goal** suite has distinct goals while keeping the objects and layout fixed; the **Object** suite contains pick-and-place tasks for different objects in the scene but with the same layout. We also create 2 *additional* task suites (from LIBERO-90): **Living Room**, and **Study Room**. We adopt 8 tasks from each of the 5 adaptation task suites, respectively. Finally, we separately evaluate each task sequentially in **LIBERO-10**, a benchmark with 10 challenging long-horizon tasks. See Fig. 3 for task suite examples and Appendix Sec. D for more details.

### 5.2  Experiment setup

**Evaluation metrics.** The primary metric we report is average per-task *suite* success rate, measured by checking if current state aligns with pre-defined goal states. For continual learning, we also assess **Forward Transfer** (FWT) and **Backward Transfer** (BWT) across the curriculum of suites. Following the metric proposed in LIBERO [48], FWT is computed by the maximum success rate one algorithm can achieve when adapting to a new task. We denote FWT at task $k$ as $\mathbf{F}_k$. Meanwhile, BWT measures the success rate increase on previous tasks. Namely, when adapting to the $k$-th task, we first record the best FWT model on this task and then evaluate this model on all previous $k-1$ tasks, obtaining success rate $\mathbf{S}_i, 1 \leq i \leq k-1$. Then we compute the success rate difference between the new model and the best FWT of the previous $k-1$ tasks and then average among them to obtain the BWT metric: $\mathbf{B} = \frac{1}{k-1}\sum_{i=i}^{k-1}(\mathbf{S}_i - \mathbf{F}_i)$. For both metrics, higher is better.

**Continual Learning Baselines.** We adopt four baselines: Full Fine-Tuning (FFT), Frozen Pretrained Feature (FPF), Experience Replay (ER) [49], and Elastic Weight Consolidation (EWC) [50]. FPF mirrors the linear probing method [42] but also tunes both the policy head and the fusion module per task. ER employs a buffer for prior task datasets, using a 50-50 data split between new and previous tasks during new task training [51]. EWC, a regularization technique, restricts the update of parameters from earlier tasks based on the Fisher information. Baseline details are in Appendix B.1.

**TAIL Adapters.** We use LoRA [16], Bottleneck Adapter [14], and Prefix Tuning [15] to represent parallel, sequential, and prefix integration styles. RoboAdapter [31], a specific implementation for decision-making, stands as another *sequential* integration style. Configuration specifics for these adapters are available in Appendix B.2. Model architectural details can be found in Appendix A.

**Training, Adaptation, and Evaluation.** Each task provides 50 high-quality human demonstrations. These are divided into 40 training trajectories and 10 for validation. *We report success rates over the validation scenes (unseen in training).* We train on and evaluate adaptation performance on all tasks within a task suite simultaneously.[1] We pretrain on **Kitchen** until performance convergence (100 epochs). Subsequent adaptations follow two setups: (1) sequential adaptation across the **Spatial**, **Goal**, **Object**, **Living Room**, and **Study Room** task suites for 100 epochs each, and (2) adaptation to each long-horizon task within the **LIBERO-10** benchmark over 50 epochs. Each experiment is conducted with 3 different random seeds.

---

[1]LIBERO [48] originally evaluated on a per-task basis.

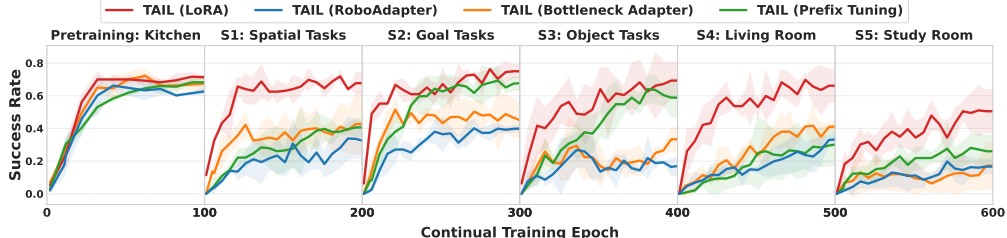

Figure 4: Success rates for different types of adapters under our TAIL framework. None of these methods suffer from catastrophic forgetting, so backward evaluation results are not presented here. LoRA performs best across all tasks, underscoring the benefits of the parallel integration approach.

In the pretraining phase, adapters are added only for the spatial and instruction encoders with CLIP weight. The GPT2 temporal encoder, fusion module, and policy head are fully tuned. During adaptation, adapters are incorporated for all encoders. Hyperparameters are presented in Appendix B.

## 5.3 Results and analysis

**Comparison of TAIL Integration Styles.** Fig. 4 showcases the continual adaptation success rates for different TAIL methods. The efficacy of LoRA suggests that a well-pretrained model has a surprisingly low intrinsic dimension for imitation learning tasks [47]. This implies the existence of a low-rank reparameterization that is just as adept for fine-tuning as the full parameter space. Further, the prefix tuning method outperforms the bottleneck-based approach [14], indicating that the sequential integration style may not be the optimal choice for continual learning, potentially due to its inherent "filtering" mechanism. Surprisingly, RoboAdapter [31] generally performs the worst, potentially due to only introducing weights after the feedforward layer as opposed to after [14] or within [15, 16] the attention layer. Due to LoRA's pronounced effectiveness, it is predominantly employed as our TAIL integration method in subsequent experiments.

Table 1: Adaptation results on 10 long horizon tasks. The ↑ symbol means the higher, the better. The BWT ↑ for TAIL methods are all 0 (no catastrophic forgetting). We highlight the best method (highest FWT ↑) in **bold**. FPF results were omitted due to its near-zero performance.

| Task | Conventional Fine-Tuning Methods | | | | | | TAIL-based Methods (**Ours**) | | | |
|---|---|---|---|---|---|---|---|---|---|---|
| | Full Fine-Tuning | | Experience Replay | | EWC | | LoRA | Prefix | Bottleneck | RoboAdapter |
| | FWT ↑ | BWT ↑ | FWT ↑ | BWT ↑ | FWT ↑ | BWT ↑ | FWT ↑ | FWT ↑ | FWT ↑ | FWT ↑ |
| Task 1 | $0.42 \pm 0.07$ | - | $0.25 \pm 0.12$ | - | $0.38 \pm 0.12$ | - | $\mathbf{0.62 \pm 0.00}$ | $0.38 \pm 0.12$ | $0.21 \pm 0.14$ | $0.12 \pm 0.00$ |
| Task 2 | $0.58 \pm 0.07$ | $-0.42 \pm 0.06$ | $0.58 \pm 0.07$ | $-0.25 \pm 0.10$ | $0.54 \pm 0.07$ | $-0.38 \pm 0.10$ | $\mathbf{0.75 \pm 0.00}$ | $0.58 \pm 0.19$ | $\mathbf{0.75 \pm 0.12}$ | $0.50 \pm 0.12$ |
| Task 3 | $0.71 \pm 0.07$ | $-0.50 \pm 0.10$ | $0.67 \pm 0.07$ | $-0.42 \pm 0.19$ | $0.38 \pm 0.12$ | $-0.46 \pm 0.12$ | $\mathbf{0.96 \pm 0.07}$ | $0.88 \pm 0.22$ | $0.71 \pm 0.19$ | $0.50 \pm 0.25$ |
| Task 4 | $\mathbf{0.96 \pm 0.07}$ | $-0.57 \pm 0.13$ | $0.92 \pm 0.07$ | $-0.50 \pm 0.20$ | $0.75 \pm 0.25$ | $-0.43 \pm 0.12$ | $0.88 \pm 0.00$ | $0.71 \pm 0.07$ | $0.71 \pm 0.19$ | $0.58 \pm 0.14$ |
| Task 5 | $0.21 \pm 0.07$ | $-0.67 \pm 0.21$ | $0.33 \pm 0.14$ | $-0.60 \pm 0.25$ | $0.17 \pm 0.19$ | $-0.50 \pm 0.18$ | $\mathbf{0.62 \pm 0.12}$ | $0.17 \pm 0.07$ | $0.25 \pm 0.00$ | $0.29 \pm 0.07$ |
| Task 6 | $\mathbf{0.83 \pm 0.19}$ | $-0.57 \pm 0.26$ | $0.71 \pm 0.19$ | $-0.55 \pm 0.25$ | $0.50 \pm 0.43$ | $-0.42 \pm 0.19$ | $0.75 \pm 0.12$ | $0.79 \pm 0.14$ | $0.75 \pm 0.00$ | $0.75 \pm 0.25$ |
| Task 7 | $0.17 \pm 0.07$ | $-0.62 \pm 0.27$ | $0.12 \pm 0.00$ | $-0.58 \pm 0.25$ | $0.04 \pm 0.07$ | $-0.44 \pm 0.24$ | $\mathbf{0.54 \pm 0.26}$ | $0.38 \pm 0.12$ | $0.31 \pm 0.09$ | $0.33 \pm 0.07$ |
| Task 8 | $0.42 \pm 0.07$ | $-0.55 \pm 0.29$ | $0.29 \pm 0.07$ | $-0.51 \pm 0.28$ | $0.12 \pm 0.18$ | $-0.46 \pm 0.28$ | $\mathbf{0.75 \pm 0.25}$ | $0.67 \pm 0.19$ | $0.25 \pm 0.18$ | $0.50 \pm 0.22$ |
| Task 9 | $0.17 \pm 0.07$ | $-0.54 \pm 0.28$ | $0.12 \pm 0.05$ | $-0.50 \pm 0.28$ | $0.00 \pm 0.00$ | $-0.41 \pm 0.29$ | $\mathbf{0.38 \pm 0.12}$ | $0.08 \pm 0.07$ | $0.19 \pm 0.09$ | $0.21 \pm 0.07$ |
| Task 10 | $0.33 \pm 0.19$ | $-0.50 \pm 0.29$ | $0.50 \pm 0.02$ | $-0.46 \pm 0.29$ | $0.12 \pm 0.18$ | $-0.38 \pm 0.31$ | $\mathbf{0.79 \pm 0.07}$ | $0.50 \pm 0.33$ | $0.44 \pm 0.09$ | $0.42 \pm 0.07$ |
| Average | $0.48 \pm 0.10$ | $-0.55 \pm 0.21$ | $0.45 \pm 0.09$ | $-0.49 \pm 0.23$ | $0.30 \pm 0.16$ | $-0.43 \pm 0.20$ | $\mathbf{0.70 \pm 0.10}$ | $0.51 \pm 0.15$ | $0.46 \pm 0.11$ | $0.42 \pm 0.13$ |

**Shortcomings of Conventional Fine-Tuning.** Across all evaluations, TAIL vastly outperforms all baselines in both forward and backward transfer, demonstrating that conventional fine-tuning methods are weak in data-scarce continual learning. In Fig. 5 we plot continual learning success rates over 6 task suites, where TAIL outperforms the best baselines by over **3x** in some comparisons and generally achieves the best success rates. We display additional results on LIBERO-10, long-horizon tasks, in Table 1. Here, TAIL again performs best, with perfect backward transfer and forward transfer capabilities significantly better than the baselines: FFT not only exhibits marked catastrophic forgetting of earlier tasks—evidenced by the decline in success rates upon transitioning to new stages—but also compromises the model's adaptability to new tasks. This decline in forward transfer is characterized by a steady descent in success rates as training progresses. Such deterioration in flexibility has been recognized in other studies as well [43, 44].

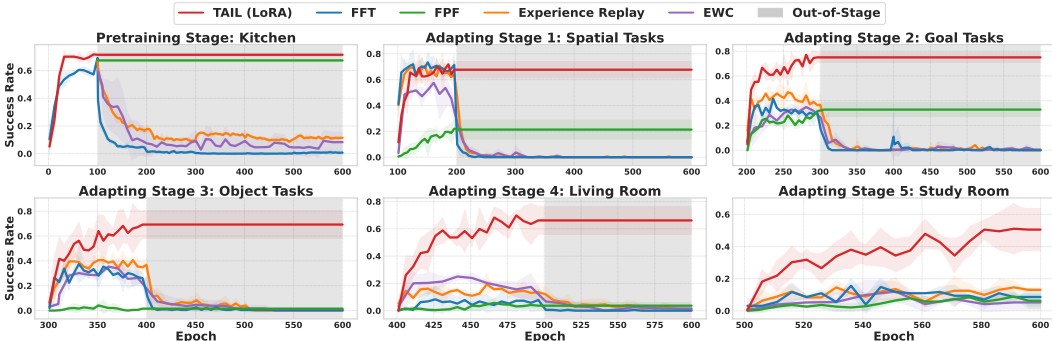

Figure 5: Success rates across 1 pretraining on 40 tasks in the Kitchen scene, and 5 adaptation stages, each with 8 tasks and 100 epochs, and will be continuously evaluated in the subsequent stages (shaded area). The start epoch for each stage is based on the order.

Table 3: Comparison of trainable parameters and memory usage for TAIL and FFT. We use (·%) and ↓ (·%) to denote the percentage of trainable parameter and the decrease of GPU memory w.r.t FFT.

| Method | Conventional | TAIL-based Methods (Ours) | | | |
| --- | --- | --- | --- | --- | --- |
| | Full Fine-Tuning | LoRA | RoboAdapter | Bottleneck Adapter | Prefix Tuning |
| CLIP (Spatial & Task Encoder) | 149.62M | 0.49M | 1.29M | 1.31M | 0.58M |
| GPT2 (Temporal Encoder) | 21.78M | 0.69M | 0.40M | 0.40M | 0.24M |
| Fusion module and policy head | 0.84M | 0.84M | 0.84M | 0.84M | 0.84M |
| Total Parameters | 172.24M | 2.02M (1.17%) | 2.53M (1.47%) | 2.55M (1.48%) | 1.66M (0.93%) |
| GPU Memory (Batch 14) | 20.1G | 15.5G (↓ 23%) | 14.0G (↓ 30%) | 14.9G (↓ 26%) | 15.8G (↓ 21%) |

265 Furthermore, exhaustive fine-tuning on specialized domains has been found to distort pretrained
266 features [42], undermining the model's adaptability. Our circle-back experiments in Table 2, wherein
267 a model fine-tuned up to Stage 5 re-trained on prior task suites, further accentuate these concerns.
268 The adaptability of FFT markedly diminishes when re-encountering prior tasks.

269 The training and validation losses, detailed in Appendix
270 C and Fig. 7, highlight FFT's propensity for overfitting.
271 This translates to a notable decline in success rates, rein-
272 forcing the challenges FFT faces in balancing retention
273 of prior tasks with the assimilation of new ones.

Table 2: The success rate of initial training and revisiting previous tasks with FFT. FFT suffers from catastrophic forgetting and performs worse on re-visits despite re-training on the same data.

274 While ER and the regularization-based method EWC
275 exhibit some potential in mitigating catastrophic forget-
276 ting, they were detrimental to forward transfer perfor-
277 mance. Their downsides are also reflected in storage

| Type | LIBERO Task Suite | | |
| --- | --- | --- | --- |
| | Spatial | Goal | Object |
| Initial | 0.79 | 0.42 | 0.42 |
| Re-visit | 0.53 (↓0.26) | 0.20 (↓0.22) | 0.27 (↓0.15) |

278 and computing costs: ER requires more storage for previous data than TAIL LoRA adapter weights
279 (e.g., Kitchen dataset at 28GB vs 7.8MB for TAIL's LoRA adapter). Furthermore, EWC presents
280 significant challenges for larger models because of the increased GPU memory consumption from
281 maintaining a copy of the entire weights of the old model in memory. We also found it to exhibit
282 unstable training due to the regularization loss. More discussions are presented in Appendix B.1.

283 **Conclusion.** In this study, we examined the challenges of efficiently adapting large pretrained models
284 for decision-making and robotics applications. We proposed TAIL, an efficient adaptation framework
285 for pretrained decision-making models. Through a comprehensive exploration of parameter-efficient
286 fine-tuning (PEFT) techniques in TAIL, especially Low-Rank Adaptation (LoRA), we demonstrated
287 their potential in enhancing adaptation efficiency, mitigating catastrophic forgetting, and ensuring
288 robust performance across diverse tasks. TAIL offers a promising avenue for the efficient adaptation of
289 large decision-making models. Despite the fact that our method requires significantly less computation
290 and memory (and storage), our experiments show that it consistently outperforms all prior approaches
291 in the continual imitation learning setting. As the demand for adaptive, intelligent agents grows
292 across various domains, the insights from this research offer a promising direction for the future of
293 efficient model adaptation in decision-making contexts.

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

# Appendix: TAIL: Task-specific adapters for imitation learning with large pretrained models

## A  Model Architecture Details

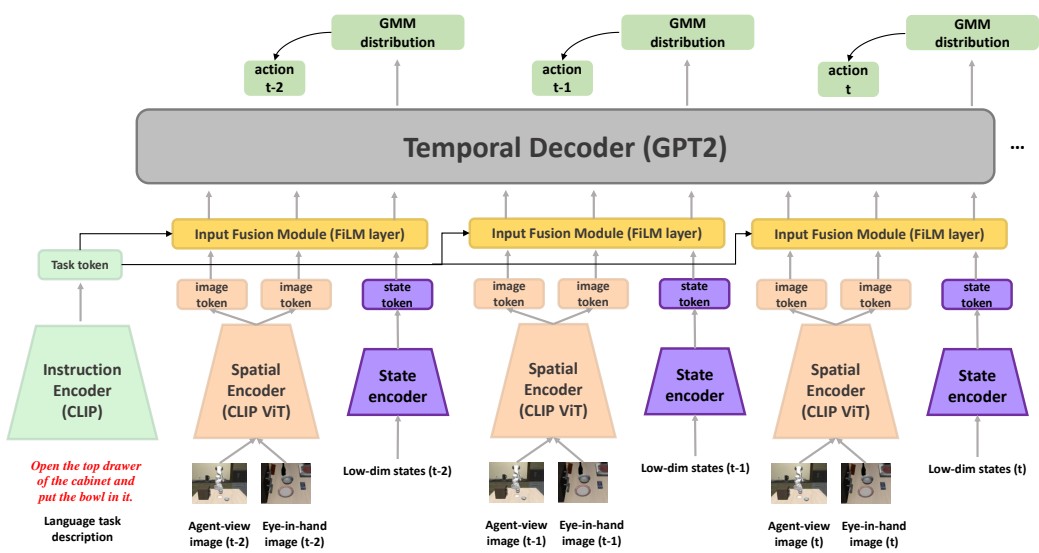

Figure 6: A detailed view of the multi-modal, transformer policy architecture we utilize for pretraining. We encode language task descriptions with a pretrained CLIP instruction encoder and image observations with a pretrained CLIP spatial encoder. We additionally encode robot state observations which, along with the observation embeddings, are embedded into a sequence of tokens used by the temporal decoder transformer to predict single-step action distributions. We include an input fusion module (FiLM [41]) to explicitly combine the task embedding with the observation and state embeddings for better instruction-following ability.

### A.1  Pretrained Input Encoders

We utilize pretrained CLIP image and textual encoders [2] to encode image observations and language goal descriptions, respectively. Note that we do not use a pre-trained encoder for the low-dimensional state; the state encoder is learned from scratch.

### A.2  Input Modality Fusion

We utilize Feature-wise Linear Modulation (FiLM) layers [41] (Fig. 1(a), input fusion module) to fuse language task specifications with image observations. FiLM is a technique in multi-modal deep learning which modulates the intermediate activations of a neural network based on external information. Rather than explicitly designing architectures for conditional computation, FiLM layers simply use the features from one network to modulate the features of another.

Let's consider a neural network $f$ with intermediate activations $x$ and an external network $g$ which outputs modulation parameters $\gamma$ and $\beta$. The modulated features $x'$ are given by:

$$\gamma, \beta = g(z) \tag{4}$$
$$x' = \gamma \odot x + \beta, \tag{5}$$

where $z$ is the input to the external network $g$; $\odot$ represents element-wise multiplication; $\gamma$ and $\beta$ are vectors having the same size as $x$, with each element modulating a corresponding feature in $x$.

Thus, FiLM layers allow for a dynamic and feature-wise conditional computation without needing explicit architectural changes. As such, task token (language) embeddings are given as input to a fully connected feedforward network, which outputs scale and translation parameters for the image and state embeddings. These parameters modulate the image and state embeddings before they are passed to the transformer backbone.

### A.3 Temporal Transformer Backbone

We utilize a standard GPT-2 [40] transformer backbone for our policy. Its input is a sequence of image and low-dim state encodings (robot joint states in LIBERO) and it outputs an action distribution. Following the literature [52, 48], we adopt a stochastic policy parametrization based on a Gaussian-Mixture-Model (GMM) [53]. Therefore, for every decision-making step, the transformer produces a latent vector of Gaussian means and variances, one for each of the GMM modes. We optimize the parameters of the model with the negative log-likelihood loss on the ground truth actions based on the parameters of the GMM distribution. At evaluation time, we deterministically select the next action parameterized by the mean of the Gaussian model with the highest density.

The environment configuration and the temporal decoder (GPT-2) hyperparameters are presented in Table 4.

Table 4: Environment configuration and GPT-2 model hyperparameters

| Environment Configuration | | GPT2 Temporal Encoder Configuration | | | |
|---|---|---|---|---|---|
| Action Dim. | 7 | Max Seq Length | 8 | Activation | Gelu New |
| Raw State Dim. | 9 | Number of Heads | 8 | Number of Layers | 6 |
| Max Episode Length | 500 | GMM Min Std | 0.0001 | GMM Modes | 5 |
| Image Resolution | 128 x 128 | FiLM Layers | 2 | Dropout | 0.15 |
| Image Views | Agent Front, Eye-in-Hand | | | | |

# B  Implementation and Training Details

## B.1  Baseline Details

**Experience Replay (ER).** ER [49, 51] is a rehearsal-based approach that retains a buffer of samples from previous tasks to facilitate the learning of new tasks. After completing the learning process for a task, a subset of the data is saved into this buffer. During the training of subsequent tasks, ER draws samples from this buffer and mixes them with current task data. This process ensures that the training data closely resembles the distribution of data across all tasks. In our setup, we store all the previous trajectories in a replay buffer. For each training iteration on a new task, we uniformly sample $50\%$ trajectories from this buffer and $50\%$ from the new task's training data, respectively.

**Elastic Weight Consolidation (EWC).** EWC [50] is a regularization method that adds a term to the standard single-task learning objective to constrain the updates of the neural network. This constraint uses the Fisher information matrix to gauge the significance of each network parameter. The loss function for task $k$ is represented as:

$$L_{\text{EWC}_k}(\theta) = L_{\text{BC}_K}(\theta) + \sum_i \frac{\lambda}{2} F_i (\theta_i - \theta_{k-1,i}^*)^2$$

Here, $\lambda$ is a hyperparameter penalty, and $F_i$ is the diagonal of the Fisher information matrix given by:

$$F_k = \mathbb{E}_{s \sim D_k, a \sim p_\theta(\cdot|s)} \left( \nabla_{\theta_k} \log p_{\theta_k}(a|s) \right)^2$$

For our experiments, we adopt the online version of EWC. It updates the Fisher information matrix using an exponential moving average throughout the lifelong learning process. The actual Fisher Information Matrix estimate used is:

$$\tilde{F}_k = \gamma F_{k-1} + (1 - \gamma) F_k$$

with $F_k = \mathbb{E}_{(s,a)\sim D_k} \left( \nabla_{\theta_k} \log p_{\theta_k}(a|s) \right)^2$ and $k$ representing the task number. Following the benchmark implementation [48], the hyperparameters are set as $\gamma = 0.9$ and $\lambda = 5 \times 10^4$.

**Discussions.** Both Experience Replay (ER) and Elastic Weight Consolidation (EWC) demonstrate potential in mitigating catastrophic forgetting. However, they each come with notable limitations, particularly with respect to forward transfer performance, storage, and computational efficiency.

*Storage Overhead:* ER demands significant storage space to maintain samples from prior tasks. This becomes particularly evident when comparing the storage needs of ER for larger datasets, such as the Kitchen dataset which requires 28GB, with the lightweight LoRA adapter occupies only 7.8MB. The vast difference in storage demands underscores the inefficiency of the ER approach.

*Computational Challenges:* EWC, by design, necessitates the maintenance of a copy of the weights of the previous model in GPU memory. This leads to escalated GPU memory consumption, making EWC tends to reduce the training batch size, subsequently slowing down the training process.

*Training Instability:* The regularization approach of EWC can introduce instability during training, owing to the regularization loss. This is also reflected by the poor forward transfer capability, as shown in Table 1.

*Scalability Concerns:* While EWC might be manageable for smaller networks, it is ill-suited for the fine-tuning of larger decision models due to its computational and storage challenges.

Given these outlined limitations, we advocate TAIL for alternative approaches that are both storage-efficient and computationally scalable, especially for large pretrained model adaptation.

## B.2 TAIL Adapter Configurations

To establish our TAIL adapter configurations, we primarily draw from the AdapterHub implementation, setup and hyperparameters [54].

We utilize the default hyperparameters for LoRA, with the rank $r = 8$ and scaling factor $\alpha = 8$. These low-rank matrices are applied in parallel to the Transformer's query and value matrices [16]. We also adopt the default for prefix token length of 30 for the prefix tuning [15] method across all tasks. To improve the training stability, Low-rank matrices ($r = 16$) are employed during training to represent the prefix tokens. The Bottleneck Adapter [14] employs the bottleneck size of 32, and is applied to both the output layer of the attention and the intermediate feedforward layers. The RoboAdapter method [31], as the closest work to us, also applies the sequential adapters to the decision-making domain. It differs from the Bottleneck Adapter in that they adopt a special insertion of weights to specific layers of the Transformer, namely, layers $0, 1, 5, 6, 10, 11$. They selectively skip certain layers, aiming to increase the bottleneck size on the remaining layers. Therefore, the bottleneck size is doubled to 64 for this approach, such that all methods share similar amount of parameters.

In order to maintain balanced adapter parameters number between the two CLIP-based (spatial and instruction) encoders, and the temporal transformer GPT2 decoder, the rank size for the GPT2 decoder is doubled across all methodologies. This adjustment compensates for the GPT2 decoder's fewer layers relative to the encoders.

For the continual learning setup, we use the previous stage's adapter weight (if any) plus a small random Gaussian noise with standard deviation 0.001 as an initialization of the current stage. The goal for adding a minor random noise aims to improve the adapter weight capacity [42, 55, 43], preventing the weights from being trapped into local optimum. There is a potential to better utilize the trained adapter weights on preceding tasks. We outline several promising exploration directions in Appendix Section B.4.

## B.3 Training Hyperparameters and Experiment Configurations

Following similar setup as in the LIBERO benchmark [48], we perform data augmentation for the image observation data for all methods. We adopt the color, affine, and random erase augmentations to improve the robustness. The hyperparameters are presented in Table 5.

Table 5: Image data augmentation and training hyperparameters

| Image Augmentation | | | | Training and Optimizer Configuration | | | |
|---|---|---|---|---|---|---|---|
| Brightness | 0.3 | Contrast | 0.3 | Training Epochs | 100/50 | Batch Size (per device) | 10/14/18 |
| Saturation | 0.3 | Hue | 0.3 | Training Epochs per Eval | 5 | Eval Episodes/Task | 8 |
| Color Aug Prob. | 0.9 | Affine Degrees | 15 | Warm-up Steps | 500 | Weight Decay | 0.1 |
| Affine Translate | 0.1 | Affine Prob. | 0.6 | Learning Rate (LR) | 1e-4 | LR Scheduler | Linear |
| Random Erase Prob. | 0.1 | | | Training Demo Num | 40 | Validation Demo Num | 40 |

For our training process, we employed the AdamW optimizer combined with a linear learning rate scheduler. The majority of our task suites—Kitchen, Spatial, Goal, Object, Living Room, and Study Room—underwent training for 100 epochs. Notably, each suite encompasses multiple tasks, with Kitchen having 40 and the others containing 8 each. In contrast, the 10 long-horizon adaptation tasks, termed LIBERO-10, were trained for 50 epochs, with each task trained sequentially. We performed evaluations after every 5 training epochs over 8 episodes (unseen in training) for each task.

**Computing machine.** Our experimental platform was powered by an AMD EPYC 7R32 CPU running Ubuntu 20.04.06. All trainings utilized 8 NVIDIA A10G GPUs, each with a memory of 22731 MiB, equipped with driver version 470.199.02 and CUDA version 11.4. We employ Distributed Data Parallel (DDP) for parallel training across 8 GPUs, and utilize the 16-bit floating point precision (FP16) training mode to accelerate the training process. To ensure reproducibility, we adopted 3 distinct random seeds: 0, 21, and 42.

**Training time.** For a holistic perspective on training duration: FFT and ER methods demanded between $120 \sim 140$ hours per experiment ($1.5 \sim 1.75$ hours per task) for the 6 task suites shown in Fig. 5, including the evaluation time. In stark contrast, TAIL-based techniques slashed this to $60 \sim 66$ hours ($0.75 \sim 0.825$ hours per task). Hence, TAIL would also be much cheaper to train, considering its significantly shorter training time under identical computing machines.

Batch sizes varied by training method. EWC employed a batch size of 10, given its added memory demands to store a distinct full parameter set. FFT and ER utilized batch sizes of 14. Owing to TAIL's more efficient memory utilization—detailed in Table 3—a larger batch size of 18 was feasible, which can maximize GPU resource usage on our machine, reducing training duration and cost.

## B.4 More Discussion and Future Directions

The TAIL framework paves the way for a myriad of research opportunities:

1. **Better Weight Allocation Method Across Layers:** An interesting question within this framework is discerning which layers, early or later, derive the most benefit from weight modifications. This can offer insights into the adaptability of neural architectures [56].

2. **Enhanced Reusability of Trained Adapters:** Exploring methods to efficiently reuse adapters from prior tasks, especially in scenarios with limited data, is a promising direction. AdapterFusion techniques [57] can be potentially useful, enabling the composition of knowledge from multiple pre-existing adapters.

3. **Building on Knowledge with Parallel Integration:** The parallel integration method, particularly with LoRA weights, offers the capability to merge trained weights back into the main model. This iterative buildup of knowledge makes the approach valuable for continual learning, allowing new adapters to capitalize on the expertise of their predecessors.

4. **Combining with Established Continual Learning Strategies:** The potential to merge the TAIL framework with existing continual learning methods, like Experience Replay and EWC, can be

585     a beneficial avenue. Such integrations can accommodate the strengths of each method, crafting
586     models that are both efficient in memory and robust against forgetting.

587 5. **Extension beyond the Imitation Learning Domain:** Taking the TAILframework into other
588     decision-making domains, such as reinforcement learning (RL), is also promising. TAIL has the
589     potential to address the model capacity loss issue in RL [55, 43]. Leveraging the TAIL framework
590     can also aid in multitask learning, meta-learning, and efficiently adapting offline-trained RL
591     models to new tasks without the necessity of vast amounts of data or extensive fine-tuning, thereby
592     potentially accelerating convergence to optimal policies.

593 The avenues above elucidate the adaptability and potential of the TAIL framework, setting the stage
594 for future research in this domain.

# C More Experiment Results

In this section, we provide additional results from our experiments. For each task, we used 40 demonstrations for training and 10 for validation. We are interested in the following question: *In scenarios where data is limited, how resilient is TAIL against overfitting compared to traditional fine-tuning methods?* To answer this, we present the training and validation loss cross the Kitchen, Spatial, Goal, Object, Living Room and Study Room task suites, each with 100 epochs, in Fig. 7.

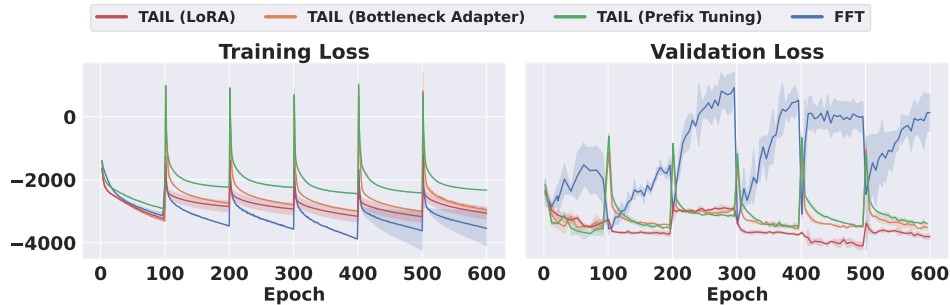

Figure 7: Adaptation loss trends: Training versus validation. The graph shows that the TAIL model consistently has more stable validation losses, which means that it is more robust to contexts with limited data. On the other hand, the full fine-tuning model (FFT) has larger validation losses, which means that it is more likely to overfit to the training data.

A noteworthy observation from Fig. 7 is the behavior of FFT. Despite achieving the lowest training loss across all stages, its validation loss spikes significantly after just a few epochs. This pattern suggests severe overfitting when FFT is applied to the entire parameter space using limited data. Intriguingly, this overfitting intensifies in the later adaptation phases, potentially signifying a distortion of pretrained features as alluded to by Kumar et al. [42]. Such distortion could be a contributor to the suboptimal success rate observed in Fig. 5, and the loss of learning capacity when revisiting a previous task, as presented in Table 2.

In contrast, TAIL-based methods shows strong resilience against overfitting. Drawing from the Occam's razor principle, TAIL leverages fewer trainable parameters, inherently reducing its potential to overfit with scarce data. Additional, different integration styles provide the flexibility to effectively utilize the features from pretrained models while preserving them across all the adaptation stages.

This observation underscores the disparities between our decision-making problem, characterized by its limited data, and the traditional language or vision domains, which have data in abundance. Prior studies utilizing parameter-efficient fine-tuning techniques for language or vision tasks often reported superior performance with full fine-tuning due to its low training loss [19, 20, 21, 31]. However, as our results demonstrate, a lower training loss does not invariably translate to superior performance, especially in the context of a data-scarce sequential decision-making tasks.

**Analysis of pretrained weights' influence**. We aim to answer the following question: *how does the underlying pretrained base model influence the performance of TAIL, and are certain pretrained weights more conducive to this kind of adaptation?* We initiated our investigation by analyzing the success rates of 40 Kitchen tasks using different pretrained weights for the spatial encoder. Apart from the CLIP-ViT pretrained encodings as we adopted in our main results, two other initialization of weights were considered: one sourced from the Visual Cortex 1 (VC-1) [58], recognized for being a leading pretrained

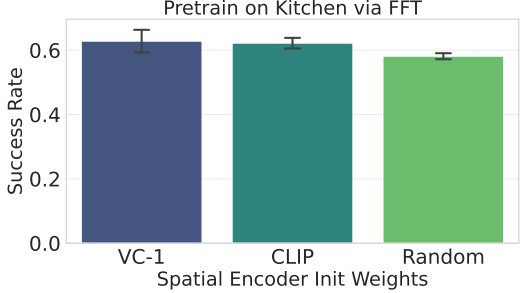

Figure 8: Training on the Kitchen task with different pretrained CLIP-ViT encoder weight. Random means using random initialization weight.

model for embodied agent tasks, and another
using randomly initialized weights. The language instruction encoder consistently utilized the CLIP
text model. From the results in Fig. 8, the VC-1 pretrained weights delivered performance on par
with the CLIP-ViT encodings. Both considerably outperformed the randomly initialized weights,
suggesting that large-scale pretraining can indeed enhance downstream fine-tuning. We then study
how does the pretrained base model influence the performance of TAIL.

**Further Evaluations on TAIL with Different Base Models.** To understand the influence of the base
model's features on the performance of TAIL, we conducted additional evaluations. In Table 6, the
methods column showcases different configurations:

- **LoRA (CLIP):** The main setup we adopted in the experiment section 5, which keeps the
  pretrained CLIP encodings frozen across all the adaptation stages.
- **LoRA (CLIP with FFT):** Starting with the CLIP model, we applied FFT pretraining on the
  Kitchen task before using LoRA for subsequent adaptations.
- **LoRA (VC-1 with FFT):** The VC-1 model, after FFT pretraining on the Kitchen task, was
  adapted using LoRA.
- **LoRA (Random with FFT):** A model with randomly initialized weights underwent FFT
  pretraining on the Kitchen task, followed by adaptation with LoRA.

All the pretrained encodings implemented in the same model architecture as described in Appendix
Section A.

Observations from Table 6 highlight several findings:

- **Dominance of Original CLIP:** The pure CLIP base model, when combined with LoRA,
  yielded the highest success rates across all task suites, suggesting the inherent quality and
  robustness of the original CLIP features for these tasks.
- **FFT's Mixed Impact:** While FFT pretraining aids in task-specific fine-tuning, when
  combined with CLIP, it leads to a degradation in performance. This could be attributed to
  FFT potentially diluting the comprehensive and rich features within CLIP [42], especially
  when exposed to a more constrained domain with limited data.
- **VC-1's Comparable Performance:** The VC-1 model, though renowned in the domain of
  embodied agent tasks, delivered results that were only marginally better than the randomly
  initialized weights when both were subjected to FFT pretraining and then adapted with
  LoRA. This emphasizes the unique advantages of the original CLIP features.

Interestingly, it is observed that CLIP is pretrained on the most comprehensive dataset, followed by
VC-1. In contrast, the model with random weights only underwent pretraining on the 40 Kitchen
tasks. The success rates mirror this order, underscoring the idea that the efficacy of TAIL is closely
tied to a base model pretrained with rich features on extensive datasets. So in summary, the choice
of base model significantly affects the performance of TAIL, with CLIP's original features showing
remarkable compatibility and resilience across various task suites

Table 6: Evaluation results of FWT for LoRA with different pretrained model weights. The higher, the better.
We highlight the best method with highest FWT as **bold**.

| Method | Spatial | Goal | Object | Living Room | Study Room | Average |
|---|---|---|---|---|---|---|
| LoRA (CLIP) | **0.76** ± 0.02 | **0.79** ± 0.02 | **0.73** ± 0.14 | **0.73** ± 0.07 | **0.55** ± 0.11 | **0.71** ± 0.07 |
| LoRA (CLIP with FFT) | 0.62 ± 0.04 | 0.67 ± 0.13 | 0.38 ± 0.08 | 0.32 ± 0.08 | 0.32 ± 0.01 | 0.46 ± 0.07 |
| LoRA (Random with FFT) | 0.38 ± 0.19 | 0.60 ± 0.06 | 0.37 ± 0.03 | 0.23 ± 0.01 | 0.47 ± nan | 0.41 ± 0.07 |
| LoRA (VC-1 with FFT) | 0.56 ± 0.07 | 0.66 ± 0.08 | 0.25 ± 0.00 | 0.20 ± 0.06 | 0.48 ± 0.07 | 0.43 ± 0.05 |

# D  Evaluation Task Details

We list all the language instruction describing the tasks we adopted in our experiments as follows
[48].

| Task Suite | Instructions |
| --- | --- |
| Kitchen | close the top drawer of the cabinet |
| | close the top drawer of the cabinet and put the black bowl on top of it |
| | put the black bowl in the top drawer of the cabinet |
| | put the butter at the back in the top drawer of the cabinet and close it |
| | put the butter at the front in the top drawer of the cabinet and close it |
| | put the chocolate pudding in the top drawer of the cabinet and close it |
| | open the bottom drawer of the cabinet |
| | open the top drawer of the cabinet |
| | open the top drawer of the cabinet and put the bowl in it |
| | put the black bowl on the plate |
| | put the black bowl on top of the cabinet |
| | open the top drawer of the cabinet |
| | put the black bowl at the back on the plate |
| | put the black bowl at the front on the plate |
| | put the middle black bowl on the plate |
| | put the middle black bowl on top of the cabinet |
| | stack the black bowl at the front on the black bowl in the middle |
| | stack the middle black bowl on the back black bowl |
| | put the frying pan on the stove |
| | put the moka pot on the stove |
| | turn on the stove |
| | turn on the stove and put the frying pan on it |
| | close the bottom drawer of the cabinet |
| | close the bottom drawer of the cabinet and open the top drawer |
| | put the black bowl in the bottom drawer of the cabinet |
| | put the black bowl on top of the cabinet |
| | put the wine bottle in the bottom drawer of the cabinet |
| | put the wine bottle on the wine rack |
| | close the top drawer of the cabinet |
| | put the black bowl in the top drawer of the cabinet |
| | put the black bowl on the plate |
| | put the black bowl on top of the cabinet |
| | put the ketchup in the top drawer of the cabinet |
| | close the microwave |
| | put the yellow and white mug to the front of the white mug |
| | open the microwave |
| | put the white bowl on the plate |
| | put the white bowl to the right of the plate |
| | put the right moka pot on the stove |
| | turn off the stove |

Table 7: 40 Kitchen scene pretraining tasks

| Task Suite | Instructions |
|---|---|
| Long-horizon (LIBERO 10) | put both the alphabet soup and the tomato sauce in the basket
put both the cream cheese box and the butter in the basket
turn on the stove and put the moka pot on it
put the black bowl in the bottom drawer of the cabinet and close it
put the white mug on the left plate and put the yellow and white mug on the right plate
pick up the book and place it in the back compartment of the caddy
put the white mug on the plate and put the chocolate pudding to the right of the plate
put both the alphabet soup and the cream cheese box in the basket
put both moka pots on the stove
put the yellow and white mug in the microwave and close it |
| Spatial | pick up the black bowl between the plate and the ramekin and place it on the plate
pick up the black bowl next to the ramekin and place it on the plate
pick up the black bowl from table center and place it on the plate
pick up the black bowl on the cookie box and place it on the plate
pick up the black bowl in the top drawer of the wooden cabinet and place it on the plate
pick up the black bowl on the ramekin and place it on the plate
pick up the black bowl next to the cookie box and place it on the plate
pick up the black bowl on the stove and place it on the plate |
| Goal | open the middle drawer of the cabinet
put the bowl on the stove
put the wine bottle on top of the cabinet
open the top drawer and put the bowl inside
put the bowl on top of the cabinet
push the plate to the front of the stove
put the cream cheese in the bowl
turn on the stove |
| Object | pick up the alphabet soup and place it in the basket
pick up the cream cheese and place it in the basket
pick up the salad dressing and place it in the basket
pick up the bbq sauce and place it in the basket
pick up the ketchup and place it in the basket
pick up the tomato sauce and place it in the basket
pick up the butter and place it in the basket
pick up the milk and place it in the basket |
| Living Room | pick up the alphabet soup and put it in the basket
pick up the butter and put it in the basket
pick up the milk and put it in the basket
pick up the orange juice and put it in the basket
pick up the tomato sauce and put it in the basket
pick up the alphabet soup and put it in the tray
pick up the butter and put it in the tray
pick up the cream cheese and put it in the tray |
| Study Room | pick up the book and place it in the right compartment of the caddy
pick up the book and place it in the front compartment of the caddy
pick up the book and place it in the left compartment of the caddy
pick up the book and place it in the right compartment of the caddy
pick up the red mug and place it to the right of the caddy
pick up the white mug and place it to the right of the caddy
pick up the book in the middle and place it on the cabinet shelf
pick up the book on the left and place it on top of the shelf |

Table 8: Adaptation task suites

