# OpenReview forum: "TAIL: Task-specific Adapters for Imitation Learning with Large Pretrained Models"
_robot-learning.org/CoRL/2023/Workshop/TGR — CoRL 2023 Workshop TGR Poster_

### Official Review · Reviewer_7QYi · 2023-10-16

**Rating:** 7
**Confidence:** 3

**Review:**

This paper proposes a method for using LLM to perform task adaption to imitation learning. This work is useful for generalizing learned models to a variety of scenarios and fits this workshop's topic.

---

### Official Review · Reviewer_a9pD · 2023-10-19

**Rating:** 8
**Confidence:** 3

**Review:**

This paper proposed TAIL (Task-specific Adapters for Imitation Learning), an efficient adaptation framework for pretrained decision-making models. The authors demonstrated that a comprehensive exploration of parameter-efficient fine-tuning (PEFT) techniques in TAIL, especially Low-Rank Adaptation (LoRA) can be extremely effective in enhancing adaptation efficiency, mitigating catastrophic forgetting, and ensuring robust performance across diverse tasks. The idea of leveraging large-scale pre-trained models and efficiently finetuning on downstream robotics task for adaptability can be highly relevant with the main focus of this workshop.

---

### Decision · Program_Chairs · 2023-10-20

**Decision:**

Accept (Poster)

**Comment:**

Great paper and closely aligned topic!